# Comparing Oncologic Outcomes and Toxicity for Combined Modality Therapy vs. Carbon-Ion Radiotherapy for Previously Irradiated Locally Recurrent Rectal Cancer

**DOI:** 10.3390/cancers15113057

**Published:** 2023-06-05

**Authors:** Elizabeth B. Jeans, Daniel K. Ebner, Hirotoshi Takiyama, Kaitlin Qualls, Danielle A. Cunningham, Mark R. Waddle, Krishan R. Jethwa, William S. Harmsen, Joleen M. Hubbard, Eric J. Dozois, Kellie L. Mathis, Hiroshi Tsuji, Kenneth W. Merrell, Christopher L. Hallemeier, Anita Mahajan, Shigeru Yamada, Robert L. Foote, Michael G. Haddock

**Affiliations:** 1Department of Radiation Oncology, Mayo Clinic, 200 2nd Street SW, Rochester, MN 55905, USA; jeans.elizabeth@mayo.edu (E.B.J.);; 2QST Hospital, National Institutes for Quantum Science and Technology, 4-9-1 Anagawa, Inageku, Chiba 263-8555, Japan; 3Department of Statistics, Mayo Clinic, 200 2nd Street SW, Rochester, MN 55905, USA; 4Division of Medical Oncology, Mayo Clinic, 200 First Street SW, Rochester, MN 55905, USA; 5Division of Colon & Rectal Surgery, Mayo Clinic, 200 2nd Street SW, Rochester, MN 55905, USA

**Keywords:** carbon-ion radiotherapy, combined modality treatment, locally recurrent rectal cancer, radioresistance, particle therapy

## Abstract

**Simple Summary:**

The efficacy of carbon-ion radiotherapy (CIRT) alone vs. multimodality therapy, including surgery, chemoradiotherapy, and intraoperative radiotherapy for previously irradiated locally recurrent rectal cancer, has not been evaluated. A total of 85 patients receiving 70.4 Gy (RBE) in 16 fractions of CIRT were compared with 86 patients receiving 30 Gy in 15 fractions of chemoradiation, resection, and intraoperative electron radiotherapy. CIRT demonstrated improved two- and five-year overall survival (83.1% and 46.8% vs. 62.5% and 25.7% for CMT), with statistically equivalent local recurrence and disease progression, with reduced treatment toxicity and decreased patient cost. A prospective comparison is warranted.

**Abstract:**

No standard treatment paradigm exists for previously irradiated locally recurrent rectal cancer (PILRRC). Carbon-ion radiotherapy (CIRT) may improve oncologic outcomes and reduce toxicity compared with combined modality therapy (CMT). Eighty-five patients treated at Institution A with CIRT alone (70.4 Gy/16 fx) and eighty-six at Institution B with CMT (30 Gy/15 fx chemoradiation, resection, intraoperative electron radiotherapy (IOERT)) between 2006 and 2019 were retrospectively compared. Overall survival (OS), pelvic re-recurrence (PR), distant metastasis (DM), or any disease progression (DP) were analyzed with the Kaplan–Meier model, with outcomes compared using the Cox proportional hazards model. Acute and late toxicities were compared, as was the 2-year cost. The median time to follow-up or death was 6.5 years. Median OS in the CIRT and CMT cohorts were 4.5 and 2.6 years, respectively (*p* ≤ 0.01). No difference was seen in the cumulative incidence of PR (*p* = 0.17), DM (*p* = 0.39), or DP (*p* = 0.19). Lower acute grade ≥ 2 skin and GI/GU toxicity and lower late grade ≥ 2 GU toxicities were associated with CIRT. Higher 2-year cumulative costs were associated with CMT. Oncologic outcomes were similar for patients treated with CIRT or CMT, although patient morbidity and cost were lower with CIRT, and CIRT was associated with longer OS. Prospective comparative studies are needed.

## 1. Introduction

Locally recurrent rectal cancer (LRRC) in previously irradiated patients is associated with poor prognosis and difficult treatment-related decisions [1,2,3,4,5,6,7,8,9,10,11,12,13,14,15,16]. Combined modality therapy (CMT) is a treatment methodology used at tertiary care centers for LRRC worldwide, including in the United States, consisting of external beam radiotherapy (EBRT), concomitant chemotherapy, surgical resection, and intraoperative electron radiotherapy (IOERT) [1,2,3,4,5,6,7,8,9,10,11,12,13,14,15,16]. In Asia and Europe, the use of carbon-ion radiotherapy in the setting of recurrent and radioresistant diseases is becoming common, though the technology is not yet available in North America [17,18,19].

For some patients, long-term disease control and survival are achievable with the use of an aggressive local therapy [1,2,3,4,5,6,7,8,9,10,11,12,13,14,15,16]. Updated modern retrospective series have highlighted promising disease outcomes with the use of CMT, though this approach comes with a substantial risk of short- and long-term morbidity. The risk of any toxicity from CMT ranges from 15–59% and most commonly includes wound complications, gastrointestinal (GI) fistulas, genitourinary toxicity (GU) (predominantly ureteral obstruction), and peripheral neuropathy [16]. Additionally, the treatment-related mortality ranges from 0 to 8% across various series [16]. Surgery often requires en bloc resection of adjacent organs, resulting in permanent morbidity. Significant heterogeneity across studies and lack of randomized evidence make quantifying improvement in oncologic outcomes with CMT difficult. However, given the risks, as well as continued grave prognosis, there has been substantial interest in improving the therapeutic ratio for patients with LRRC who are eligible or wish to undergo definitive curative-intent treatment.

Carbon-ion radiotherapy (CIRT), in comparison with conventional EBRT, allows the delivery of a higher biological dose to target disease with improved sparing of surrounding organs at risk (OARs), with an associated potential reduction in treatment-associated toxicity. CIRT is of particular interest given the radioresistance of LRRC, as its higher linear energy transfer (LET) compared with other forms of radiotherapy results in higher relative biological effectiveness (RBE) and consequent cell death [20].

Given the lack of comparative data, a retrospective cross-institutional comparison of oncologic outcomes, toxicity, and cost in patients with non-metastatic LRRC undergoing curative-intent treatment of either CIRT or CMT is presented.

## 2. Materials and Methods

After institutional review board approval, a data use agreement was developed between Institution A and Institution B. Because the earliest patient in the CIRT cohort was treated in 2006, all patients who received definitive treatment per institutional paradigm for previously irradiated locally confined non-metastatic recurrent disease between 2006 and 2019 were identified.

Eligible patients treated at Institution A received definitive CIRT: 7040 cGy (relative biological effect, RBE) delivered in 16 fractions without concurrent chemotherapy [18,21]. The modified Microdosimetric Kinetic Model was employed for dose calculation. This was delivered four days per week over four weeks. All patients were reviewed in clinical conference and offered CIRT if deemed unresectable or potentially unresectable, the latter of which included patients declining surgery due to anticipated ADL (Activities of Daily Living) loss due to combined bony resection or rejecting colostomy. Spacer insertions were considered and included either greater omentum or PTFE (Polytetrafluoroethylene) sheets if needed. A gross tumor volume (GTV) was determined by MRI, CT, and, if available, PET-CT. A 5 mm margin was set to clinical target volume (CTV); no elective area was treated. Ninety-five percent of the GTV was prescribed to receive 95% of the dose (V_95_ > 95%). Figure 1a demonstrates a treatment plan for a CIRT patient. In total, 85 CIRT patients were identified. Post-CIRT follow-up consisted of CT or PET scan with tumor marker lab work every three months; PR was considered for repeat CIRT, with DM treated with resection or chemotherapy as appropriate.

Eligible patients treated at Institution B underwent CMT: neoadjuvant chemoradiotherapy (3000 cGy in 15 fractions) followed by immediate surgery and IOERT [5,22,23]. The GTV was determined by MRI, PET-CT, and CT with a 5–10 mm expansion to CTV without extension into the small bowel. A 5 mm PTV expansion was used to prescribe V_95_ > 95%. A total of 86 CMT patients were identified, and Figure 1b demonstrates a sample radiotherapeutic plan. Post-treatment follow-up was conducted in three-month intervals with CT or PET imaging; salvage was considered on a case-by-case basis. Retrospective collection of patient, disease, and treatment characteristics, as well as oncologic follow-up and toxicities, was completed for all patients in both cohorts.

Descriptive median statistics (interquartile range (IQR)) were calculated for patient and treatment-related continuous variables. Numerical counts and percentages were calculated for patient and treatment-related categorical variables. The Chi-squared and Wilcoxon rank sum tests were used to compare patient characteristics. Outcomes for CMT patients were calculated from time of surgery and coincident IOERT delivery, while those for CIRT patients were calculated from time of completion of CIRT. Overall survival (OS) was estimated using the Kaplan–Meier method, and the competing risk model was used to report the cumulative incidence of pelvic recurrence (PR), distant metastasis (DM), and any disease progression. PR was defined as any recurrence within the true pelvis, incorporating both local and regional recurrences. Bone was included as a pelvic recurrence if the original recurrent disease extended to involve pelvic bone. Otherwise, this was denoted as a distant disease recurrence. Para-aortic lymphadenopathy and peritoneal recurrences were both classified as distant disease recurrences. A univariate Cox model for outcomes was utilized to determine a hazard ratio and *p*-value in outcomes between treatment modalities.

Toxicities at least possibly attributable to CMT or CIRT were analyzed. Toxicities were scored utilizing Common Terminology Criteria for Adverse Events (CTCAE) version 4.03 for both cohorts, with additional use of a dedicated IOERT toxicity scale for the CMT patients [24]. Acute and late toxicities were defined as less than or greater than 90 days, respectively. Acute toxicities were assessed as binary variables, using logistic regression to generate a *p*-value. Late toxicities were assessed using cumulative incidence estimates, treating death as a competing risk. A Cox model for assessment between treatment type and late toxicity was calculated. For late toxicities, the assumption of time starting at 90 days post-therapeutic intervention was set.

For cost comparison, standardized Medicare CMT costs (30 days prior to and 2 years after re-irradiation) were obtained for patients from an Institution A-specific Cost Data Warehouse. All costs, including evaluation and management, procedural, imaging, pharmacy, tests, labs, chemotherapy, radiotherapy, hospitalizations, and emergency department visits, were combined into a single cost and inflation-adjusted to the value of the US Dollar in 2017. For CIRT patients, radiation episode reimbursement rates were obtained from the Japanese Ministry of Health and converted to US Dollars at an exchange rate of 104.17 Japanese Yen to United States Dollars for comparison.

Data were analyzed using SAS Version 9.4, and a generated *p*-value of ≤0.05 was chosen for statistical significance.

## 3. Results

### 3.1. CMT Patients

A total of 86 CMT patients were identified; patient characteristics are shown in Table 1. All patients had undergone preoperative radiotherapy at initial diagnosis, delivering 5040 cGy in 28 fractions to a standard rectal field followed by total mesorectal excision. At the time of recurrence, all patients underwent re-irradiation followed by surgical resection and IOERT for negative (*n* = 42, 48.8%), microscopic (*n* = 32, 37.2%), or gross (*n* = 12, 14.0%) margins at a median time of three days after completion of EBRT. A median of 1500 cGy IOERT (1250–1500) was delivered using 9 MeV electrons.

### 3.2. CIRT Patients

A total of 85 CIRT patients were identified; patient characteristics are presented in Table 1. At the time of recurrence, all patients were considered unresectable and therefore referred for CIRT. Thirty-four patients underwent conventional radiotherapy at initial diagnosis (twenty-five preoperative, nine postoperative for close (*n* = 6) or positive margins (*n* = 3)). All but three patients received concurrent radiosensitizing chemotherapy. The median dose of preoperative EBRT was 4500 cGy (4320–5040) delivered in 180 cGy per fraction (180–180), and the median dose of postoperative EBRT was 5040 cGy (4500–6000) delivered in 200 cGy per fraction (180–200).

The other 50 patients received photon radiotherapy at initial recurrence. A total of 42 of 50 received concurrent radiosensitizing chemotherapy at initial recurrence. The median EBRT dose was 5040 cGy (4500–6000) delivered in 180 cGy per fraction (180–200). Furthermore, 17 of 50 patients received additional surgical intervention at recurrence, with 10 receiving sequenced preoperative RT and 7 receiving postoperative RT for residual disease. One patient received two prior courses of EBRT prior to presentation for CIRT.

The median time to recurrence prior to CIRT delivery, defined as the time from completion of the last radiotherapeutic or surgical intervention, whichever was later, was 1.68 years (1.13–2.85). No patients received concurrent chemotherapy while undergoing CIRT. Fifteen cases included spacer insertion. Following treatment, all patients with subsequent LR were considered for salvage with repeat CIRT—totaling nine patients to date—while those with DM were referred for resection or chemotherapy as warranted.

### 3.3. Outcomes

There were no differences in patient cohort characteristics except for receipt of concurrent chemotherapy during re-irradiation for the CMT cohort. Patients were followed until death or for a median of 6.5 years in survivors (CIRT 4.1 years; CMT 7.7 years).

There were 35 deaths in CIRT patients and 61 deaths in CMT patients. Median OS in the CIRT and CMT groups were 4.5 years and 2.6 years, respectively. Two- and five-year OS were 83.1% (95% CI 75.0–92.0) and 46.8% (35.2–62.3) for CIRT patients and 62.5% (52.5–74.4) and 25.7% (17.4–38.1) in CMT patients (Figure 2). On Cox modeling, CIRT was associated with a higher OS (HR 0.50 (0.33–0.76), *p* < 0.01).

There were 41 PRs in CIRT patients and 28 PRs in CMT patients. Median time to PR in the CIRT and CMT groups were 3.6 and 2.7 years, respectively. The two- and five-year cumulative incidence of PR was 41.8% (32.2–54.4) and 53.6% (43.1–66.7) for CIRT patients and 26.7% (18.5–38.5) and 36.7% (27.2–49.3) for CMT patients (Figure 3). On Cox modeling, treatment modality (CIRT vs. CMT) did not impact PR (HR 1.40 (0.87–2.27), *p* = 0.17).

There were 49 DMs in CIRT patients and 39 DMs in CMT patients. Median time to DM in the CIRT and CMT groups were 1.8 years and 2.2 years, respectively. The two- and five-year cumulative incidence of DM was 53.8% (43.8–66.1) and 61.4% (51.2–73.5) for CIRT patients and 43.4% (33.6–56.0) and 50.6% (40.4–63.2) for CMT patients (Figure 3). On Cox modeling, treatment modality (CIRT vs. CMT) did not impact DM (HR 1.20 (0.79–1.83), *p* = 0.39).

There were 62 CIRT patients and 50 CMT patients with any disease progression. The median times for any disease progression in the CIRT and CMT groups were 1.1 years and 2.0 years, respectively. The two- and five-year cumulative incidence of any disease progression was 69.3% (59.7–80.3) and 76.8% (67.7–87.0) for CIRT patients and 49.5% (39.5–61.9) and 64.0% (55.0–76.6) for CMT patients (Figure 3). On Cox modeling, treatment modality (CIRT vs. CMT) did not impact DM (HR 1.28 (0.88–1.86), *p* = 0.19).

### 3.4. Toxicity

Table 2 shows comparative odds ratios of acute (defined as less than 90 days) grade 2 or greater and grade 3 or greater gastrointestinal (GI), genitourinary (GU), skin, and nerve toxicities. Grade ≥ 2 GI, GU, and nerve toxicity rates were similar, while CMT was associated with a higher rate of grade ≥ 2 skin and grade ≥ 3 GI and GU toxicity.

Table 3 shows the cumulative incidence and hazard ratio of late (after 90 days) grade 2 or greater gastrointestinal (GI) and genitourinary (GU) toxicities. Median follow-up for late toxicity was 2.6 years (CMT 2.3 years; CIRT 2.8 years). For CMT, there were ten grade 2 or greater and seven grade 3 or greater GI toxicities. For CIRT, there were 15 grade 2 or greater and 11 grade 3 or greater GI toxicities. For CMT, there were 13 grade 2 or greater and 11 grade 3 or greater GU toxicities. For CIRT, there were four grade 2 or greater and zero grade 3 or greater GU toxicities. There was no acute or late grade 5 toxicity with either treatment modality.

### 3.5. Cost Comparison

Two-year standardized Medicare costs were available for nine patients treated with CMT in our exploratory cost comparison. The two-year standardized Medicare average and median costs were USD 133,879 (SD USD 94,115) and USD 96,499, of which the majority were related to hospital charges (average USD 76,922 and SD USD 69,924; median USD 45,470) and procedure charges (average USD 20,077 and SD USD 11,020; median USD 18,413). Radiation charges were an average of USD 10,694 (SD USD 6579) and a median of USD 8991. The radiation-specific CIRT costs are standardized at USD 30,144, with total direct costs of USD 46,118.

## 4. Discussion

In this study, there was no difference seen in PR, DM, or any disease progression. Both cohorts demonstrate a high risk of metastatic disease, calling for better methods of detecting micrometastatic disease at recurrence and improved novel systemic treatment options. CIRT was associated with a longer OS. Given no correlation to oncologic outcomes, this may be attributable to a difference in underlying patient health and comorbidities or unknown underlying biologic correlates. CIRT was associated with a lower risk of acute grade ≥ 2 skin and grade ≥ 3 GI and GU toxicities, as well as a lower risk of late grade ≥ 2 GU toxicities. CIRT was found to be a more cost-effective treatment option in comparison to CMT, owing in part to requisite hospitalization costs secondary to surgical treatment in CMT. There are no accruing randomized studies analyzing CIRT versus CMT in the setting of previously irradiated LRRC. This study is the first available to compare the outcomes of these two treatments.

Both cohorts of patients underwent care for non-metastatic LRRC at specialized institutions with aggressive locoregional treatment per institutional standards. In Europe and the United States, the use of CMT has been established through retrospective series demonstrating improved oncologic outcomes, with a proportion of patients achieving long-term survival [3,4,5,6,7,8,9,10,11,12,13,14,15,16]. From a feasibility perspective, the use of IOERT allows for dose-escalation in the setting of previous irradiation to preferentially spare radiosensitive small bowel. Comparably, at Institution A, CIRT is used for the treatment of radioresistant tumors such as LRRC, with reports demonstrating favorable oncologic outcomes across a range of diseases [18,21]. The radiobiologic advantage of CIRT is driven by higher inherent LET and consequent RBE benefit, inducing increased non-cell-cycle and non-oxygen-dependent double-strand breaks [20], with theorized potentiation of long-term local oncologic control. The physical benefit of sharp lateral penumbra and Bragg peak sparing surrounding OARs could lead to potentially reduced acute and late toxicity in comparison with CMT. Direct dose comparison between CMT and CIRT is complex, owing both to the biologic dose modeling inherent to CIRT delivery, challenges adapting the traditional alpha/beta model to both recurrent rectal tumors vs. surrounding rectal tissue and the inherent radiobiology of CIRT, as well as the typical four-day-per-week chronicity of CIRT treatment. An approximation of this using an alpha/beta ratio of approximately 3.9 [25], an EQD2 comparison yields 150 Gy (RBE, EQD2) for the 70.4 Gy (RBE) delivered in 16 fractions (BED = 99 Gy (RBE)), compared with a combined BED of 97 Gy to 133 Gy depending on if CMT patients received prior short- and long-course radiotherapy.

Patients presenting to Institution A were all considered unresectable; notably, patients with PR following CIRT were all eligible for reCIRT due to the dosimetric advantages described above, a treatment pattern that may contribute to the observed survival benefit in comparison with CMT. Nine patients have received this treatment from this cohort on a clinical trial, and these results are anticipated once trial enrollment is complete. Within the literature, these advantages have yielded a reduced risk of secondary malignancy in CIRT vs. conventional radiotherapy [26], while animal studies have suggested that the increased RBE contributes to an enhanced antimetastatic effect [27]. Given the possibility of improving outcomes while minimizing toxicity with repeated oncologic interventions, expanding access to CIRT has been of great interest.

Multiple phase I and phase II studies have investigated the use of CIRT in the setting of LRRC. Two phase I/II studies have analyzed delivering over 7000 cGy (RBE 3.0) with favorable outcomes and few toxicities; however, no patients in these series underwent prior radiotherapy [21,28]. In the Japan Carbon-ion Radiation Oncology Study Group (J-CROS) multi-institution retrospective series, 224 LRRC underwent CIRT, delivering 7040 cGy (RBE 3.0) or 7360 cGy (RBE3.0) in 16 fractions [19]. Patients with previous radiotherapy were eligible for inclusion; however, only three were included in the analysis (previous median dose 5040 cGy).

The only available study of CIRT in the setting of reirradiation is the PANDORA study at the Heidelberg Ion Beam Therapy Center (HIT). This study analyzed dose escalation using CIRT in recurrent or inoperable rectal cancer for previously irradiated patients (photon radiotherapy, median 5040 cGy) [29]. A total of 19 patients were treated with a median dose of 3600 cGy (RBE 3.0) (3600–5100 cGy). At a median follow-up of 7.8 months, four of nineteen developed local recurrences, and three developed distant diseases. No grade 3 or higher toxicities were seen. A longer follow-up has not been published. One study has compared CIRT to photon radiotherapy alone [30]. On comparing 35 CIRT and 31 photon patients, CIRT demonstrated improved local control, overall survival, and lower severe late toxicity.

This paper is the first to compare CIRT with CMT. Patient cohort characteristics were similar, and while no difference was seen in oncologic outcomes, CIRT was associated with lower acute and late toxicity, suggesting a benefit to CIRT. In an exploratory cost comparison, CMT standardized Medicare costs were high, at a median cost of USD 96,499, compared to Japanese CIRT charges of USD 30,144. A comparable longitudinal analysis was conducted with patients undergoing CIRT at the National Institute of Radiological Science (now National Institutes of Quantum Science and Technology) and multimodality therapy consisting of chemotherapy, radiotherapy, and hyperthermia at Gunma University Hospital [31]. Calculation of all direct costs demonstrated an average global cost for CIRT of USD 46,118, a finding that compares favorably to the global costs of CMT found in this study and suggests, as found in the prior study, an incremental cost-effectiveness with the use of CIRT. This analysis is rudimentary, and further study is needed, particularly as the United States advances efforts to develop a CIRT facility.

There are limitations to this retrospective data analysis. Due to the relative rarity of this clinical situation, patient cohorts are small, which limits the ability to match patients. However, a comparison of disease characteristics demonstrates similarity at the time of recurrence. Two-thirds of patients receiving CIRT represent third relapses, which biologically could differ from the secondary relapses seen in the CMT cohort. CMT patient tumors were comparably larger (median 5.0 vs. 2.9 cm), though this was not statistically significant. Additionally, in the CMT cohort, 17 patients did not have post-90-day toxicity data available, which could significantly underpower any late toxicity difference. Cost data was only available for nine CMT patients, and the CIRT charge reflects the institutional fee for CIRT without the inclusion of any ancillary costs, therefore limiting nuanced analysis. An arbitrary analysis time of two years was established for cost comparison, with observed differences in exchange rate similarly impacting comparison. Lastly, this study is retrospective, and a prospective study is warranted to investigate the true safety, efficacy, and cost of CIRT versus CMT in previously irradiated LRRC patients.

## 5. Conclusions

A comparative cross-institutional cohort study demonstrates similar rates of PR, DM, and any disease progression in patients undergoing CIRT or CMT for LRRC. The risk of DM remains extremely high in these patients and prompts the need for further trials into novel systemic agents and a need to detect micrometastatic disease earlier. CIRT is associated with a lower rate of acute and late toxicity in patients with previously irradiated LRRC patients. A randomized study evaluating oncologic outcomes, toxicity, quality of life, functional outcomes, and cost of CIRT versus CMT in the recurrent irradiated rectal cancer patient is warranted.

## Figures and Tables

**Figure 1 cancers-15-03057-f001:**
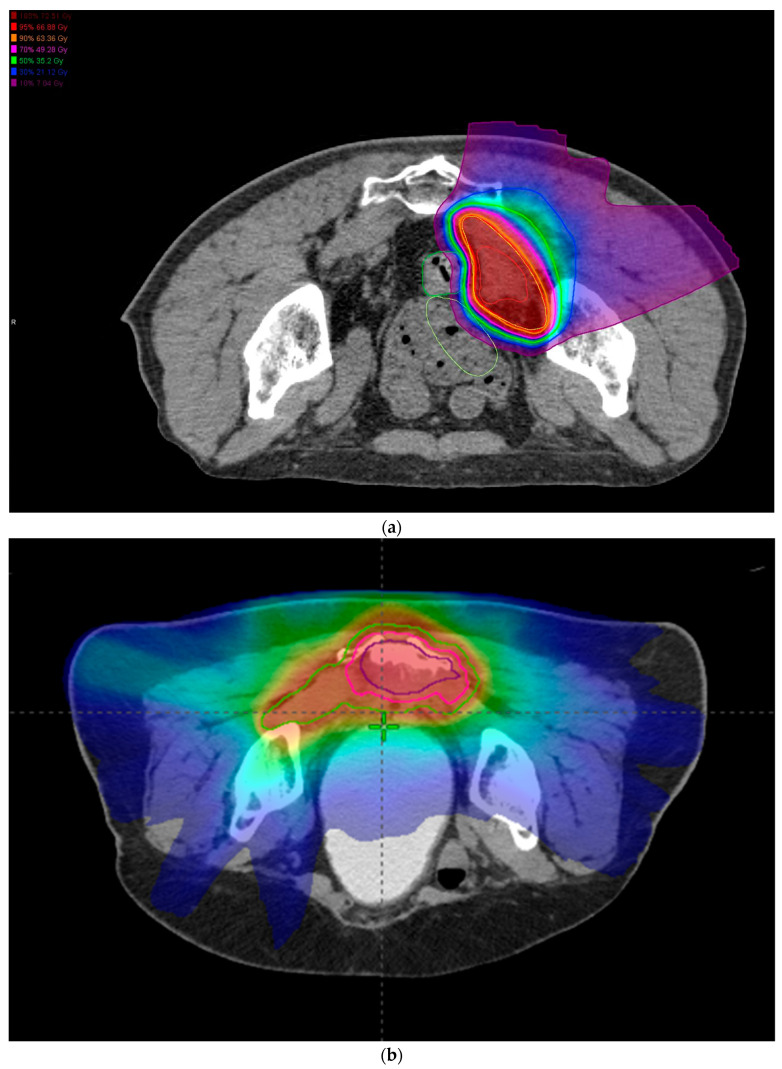
An example CIRT plan (**a**) and a CMT EBRT plan (**b**).

**Figure 2 cancers-15-03057-f002:**
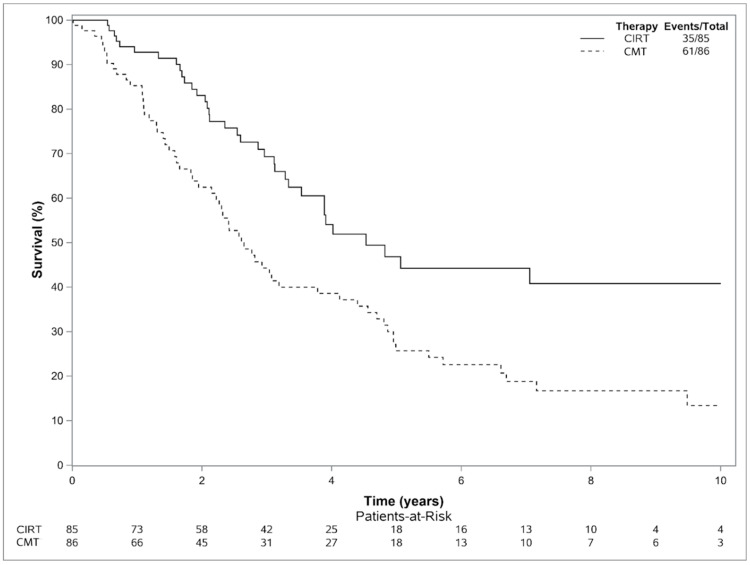
Kaplan–Meier Estimate for Death of Patients Undergoing CIRT versus CMT. On Cox modeling, CIRT demonstrated an improved OS (HR 0.50 (0.33–0.76), *p* < 0.01).

**Figure 3 cancers-15-03057-f003:**
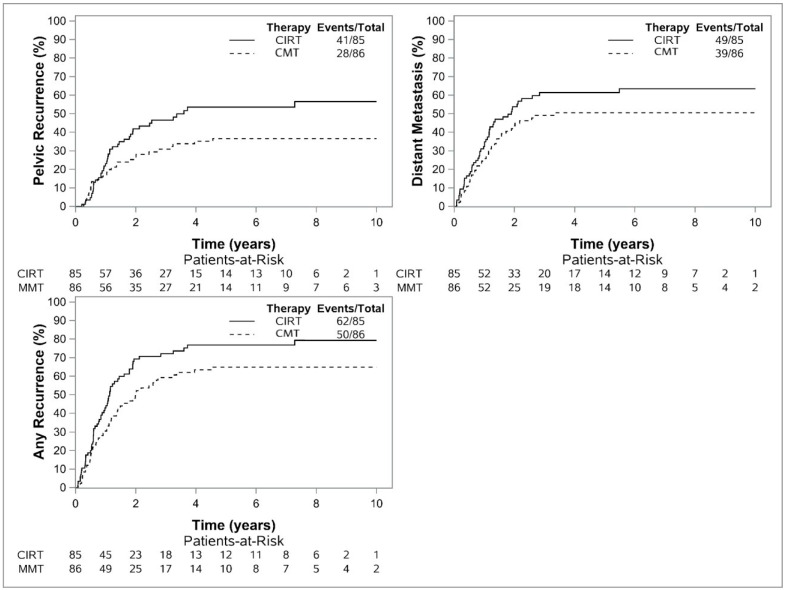
Kaplan–Meier Estimate for Pelvic Recurrence (**top left**) (HR 1.40 (0.87–2.27), *p* = 0.17), Distant Metastasis (**top right**) (HR 1.20 (0.79–1.83), *p* = 0.39), and Any Disease Progression (**bottom left**) (HR 1.28 (0.88–1.86), *p* = 0.19) for Patients Undergoing CIRT versus CMT.

**Table 1 cancers-15-03057-t001:** Patient, Recurrent Disease, and Treatment Characteristics.

	CIRT (N = 85)	CMT (N = 86)	*p*-Value
Sex (*n*, %)			NS
Female	26 (30.6%)	31 (36.0%)	
Male	59 (69.4%)	55 (64.0%)	
Age At Recurrence			NS
Median	63.0	55.1	
Q1, Q3	54.0, 68.0	48.6, 66.4	
Lymph Node Status (*n*, %)			NS
Missing	2	0	
(−)	54 (65.1%)	70 (81.4%)	
(+)	29 (34.9%)	16 (18.6%)	
Concurrent Chemotherapy (*n*, %)			*p* < 0.01
No	85 (100.0%)	4 (4.7%)	
Yes	0 (0%)	82 (95.3%)	
Year of RT Delivery, (*n*, %)			NS
2006	3 (3.5%)	6 (7.0%)	
2007	2 (2.4%)	6 (7.0%)	
2008	6 (7.1%)	6 (7.0%)	
2009	5 (5.9%)	6 (7.0%)	
2010	4 (4.7%)	16 (18.6%)	
2011	9 (10.6%)	9 (10.5%)	
2012	7 (8.2%)	6 (7.0%)	
2013	7 (8.2%)	13 (15.1%)	
2014	6 (7.1%)	8 (9.3%)	
2015	7 (8.2%)	6 (7.0%)	
2016	9 (10.6%)	1 (1.2%)	
2017	4 (4.7%)	3 (3.5%)	
2018	8 (9.4%)	0 (0.0%)	
2019	8 (9.4%)	0 (0.0%)	
Recurrence Size (cm)			NS
Median	2.9	5.0	
Q1, Q3	2.0, 4.5	3.5, 6.5	
Chemotherapy Regimens (CMT Only)			NS
5FU		38	
Capecitabine		33	
Capecitabine, Irinotecan, FOLFOX		2	
Oxaliplatin, 5FU		1	
Irinotecan, Oxaliplatin, Capecitabine		1	
Capecitabine, Camptosar		1	
5FU, FOLFOX		1	
Capecitabine, FOLFOX		1	
Leucovorin, 5FU		1	
None		7	

**Table 2 cancers-15-03057-t002:** Comparative odds ratio (95% CI) of acute (defined as less than 90 days) Grade 2+ Gastrointestinal (GI), Genitourinary (GU), Skin, and Nerve Toxicities.

	Acute Toxicity Odds Ratio (95% Confidence Interval)	
Technique	CMT vs. CIRT (Ref)	*p*-Value
Grade ≥ 2 GI	2.23 (0.79–6.24)	0.13
Grade ≥ 3 GI	21.81 (1.23–387.55)	0.04
Grade ≥ 2 GU	2.74 (0.82–9.10)	0.10
Grade ≥ 3 GU	12.76 (1.60–100.41)	0.02
Grade ≥ 2 Skin	8.06 (3.59–18.12)	<0.01
Grade ≥ 3 Skin	3.07 (1.37–6.89)	<0.01
Grade ≥ 2 Nerve	0.79 (0.31–2.01)	0.62
Grade ≥ 3 Nerve	3.00 (0.12–76.12)	0.51

**Table 3 cancers-15-03057-t003:** Cumulative Incidence and Hazard Ratio of Late (after 90 days) Grade ≥ 2 Gastrointestinal (GI) and Genitourinary (GU) Toxicities.

	Late Toxicity Cumulative Incidence (%) (95% Confidence Interval)			
Technique	Grade ≥ 2 GI	Grade ≥ 3 GI	Grade ≥ 2 GU	Grade ≥ 3 GU
CMT 1-year 2-year 3-year	11.9 (6.2–22.9) 13.5 (7.4–24.9) 15.2 (8.6–26.9)	8.9 (4.1–19.0) 8.9 (4.1–19.0) 10.5 (5.2–21.3)	10.3 (5.1–20.7) 17.0 (9.9–29.1) 18.7 (11.2–31.2)	8.8 (4.1–18.9) 13.8 (7.5–25.4) 15.5 (8.8–27.5)
CIRT 1-year 2-year 3-year	10.7 (5.7–19.8) 13.5 (7.8–23.4) 16.7 (10.1–27.7)	9.5 (4.9–18.4) 10.9 (5.9–20.2) 14.1 (8.1–24.6)	1.2 (0.2–9.7) 4.1 (1.4–12.5) 4.1 (1.4–12.5)	- - -
Hazard Ratio (95% CI) CIRT (ref) *p*-value	0.85 (0.38–1.89) 0.68	0.82 (0.32–2.11) 0.68	4.78 (1.56–14.68) <0.01	33.04 (1.71–638.40) 0.02

## Data Availability

Data supporting the reported results are maintained in a research folder in the Department of Radiation Oncology drive, Mayo Clinic, Rochester, MN, USA.

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
