# Peer review of "Comparing Oncologic Outcomes and Toxicity for Combined Modality Therapy vs. Carbon-Ion Radiotherapy for Previously Irradiated Locally Recurrent Rectal Cancer"

_cancers, 2023, doi:10.3390/cancers15113057_

Round 1
Reviewer 1 Report
This is e retrospective study on cross-institutional comparison of oncologic outcomes, toxicity and cost in patients with LRRC, previously treated with radiation therapy, unergoing to curative-intent treatment with CIRT or to more conventional CMT. This is a very relevant clinical issue because of the challange of the potential re-irradiation programs and the emerging interest in the innovative CIRT for this unfavourable tumor presentation.
The patient/tumor characteristics were similar in the cohorts of patients considered in the two institutions and while no difference was seen in some oncologic outcomes, CIRT was associated with longer OS, lower acute and late toxicity, and lower cost, suggesting a benefit to CIRT.
There are some questions need to be answered
1. Patients in cohort A (CIRT) were considered unresectable before CRT, while patients in cohort B (CMT) underwent preop RT 30Gy, resection and IORT. Had these patients resectable or potentially resectable diusease? This should be specified (potential bias in selection) (page 2 line 86 and page 3 line 95)
2. Were spacer implantation or omental flap used to create adequate distance between the the tumor (GTV) and the bowel? This should be specified, in particular for CIRT (page 2, line 89)
3. All patients with LR after CIRT were considered for re-CIRT as salvage attempt. How many patients had re-CIRT? Which was its impact on local control? This could support the observed suvival benefit of CIRT in comparison with CMT and need to be comment further (page 10, line 258)
Minor comment
there is a mistake in the name of Institution; Institution A for CIRT, not Institution B as reported (page 10, line 249 and line 256)
no comment
Reviewer 2 Report
The authors reported on a pooled retrospective analysis on Oncologic Outcomes and Toxicity for Combined Modality Therapy vs. Carbon Ion Radiotherapy for Previously Irradiated Locally Recurrent Rectal Cancer. A total of 171 previously irradiated patients was analyzed, which represents quite an impressive sample size. The content is of some interest to the readers, because it deals with a not sufficiently resolved oncological problem. The relevant literature was discussed und the presentation of the results was mainly empirical and all in all scientifically sound. Languauge was fine as expected from an English native speaking group.
Here are some suggestions for further improvement:
1. Simple summary: the abbreviation CIRT was not explained. Please use the full term with the abbreviation in brackets when first mentioning.
2. A biological comparison of the two radiotherapy dose concepts is lacking. Please discuss the differences with biological parameters such as BED or EQD2Gy or similar. How much higher was the biological CIRT dose?
3. Figure 1 looks nice but offers no real gain. Better illustration possible?
4. No details on the cocurrent chemotherapy in the CMT arm is given. Which drugs? Dose fully administered as prescribed
5. Although all patients in the CIRT arm were classified as non-resectable they had smaller tumors (not significant) than in the neoadjuvant CMT arm. That is hard to understand. How were the patients with resectable recurrences treated at the CIRT-institution? Is there a surgery department? Please clarify.
6. Figure 3: Please use the same abbreviations throughout the text: CMT instead of MMT
7. Figure 3: The curves are interesting. Beside the non-significant p-values a trend may be discussed in favor of CMT. This is interesting because of the (non-significant :-) differences in tumor size.
8. Table 3: Format difficult to read. Maybe there is a better way to visualize the comparison. some formatting necessary.
9. all data and conclusions regarding cost-effectiveness should be mitigated because of the weak data base.
